



# Climate and ablation observations from automatic ablation and weather stations at A. P. Olsen Ice Cap transect, NE Greenland, May 2008 through May 2022

Signe Hillerup Larsen[1], Daniel Binder[2,3], Anja Rutishauser[1], Bernhard Hynek[3,4], Robert Fausto[1], and Michele Citterio[1]

[1]The Geological Survey of Denmark and Greenland, Øster Voldgade 10, 1350 Copenhagen K, Denmark
[2]Institute for Geosciences, Potsdam University, Potsdam, Germany
[3]Austrian Polar Research Institute (APRI), Vienna, Austria
[4]GeoSphere Austria, Department Climate Impact Research, Vienna, Austria

**Correspondence:** Signe Hillerup Larsen (shl@geus.dk)

**Abstract.** The negative surface mass balance of glaciers and ice caps under a warming climate impacts local ecosystems, influencing the volume and timing of water flow in local catchments, while also contributing to global sea level rise. Peripheral glaciers distinct to the Greenland ice sheet respond faster to climate change than the main ice sheet. Accurate assessment of surface mass balance depends on in-situ observations of near-surface climate and ice ablation, but very few in-situ observations

of near-surface climate and ice ablation are freely available for Greenland's peripheral glaciers. The transect of three automated weather and ablation stations on the peripheral A. P. Olsen ice cap in northeast Greenland is an example of this much needed data. The transect has been monitored since 2008, and in 2022 the old weather and ablation stations were replaced by a new standardized setup. In order to ensure comparable data quality from the old and new monitoring station setups, it was necessary to re-evaluate the data collected between 2008 and 2022. This paper presents the fully reprocessed near-surface

climate and ablation data from the A. P. Olsen ice cap transect from 2008 to 2022, with a focus on data quality and the usability for ice ablation process studies. The usability and some quality issues are exemplified by using the data in an energy balance melt model for two different years. We showed that the inherent uncertainties of the data resulted in an accurate reproduction of ice ablation for just one of the two years. A transect of three automatic ablation and weather stations of this length is unique for Greenland's peripheral glaciers and it has a broad scale of usage from input to climate reanalysis to detailed

surface ablation studies. The dataset can be downloaded here: https://doi.org/10.22008/FK2/X9X9GN (Larsen and Citterio, 2023). Future refinements will be uploaded as new versions and the continuation of the transect time series are available via https://doi.org/10.22008/FK2/IW73UU (How et al., 2022).

## 1 Introduction

Under the influence of the current warming climate, glaciers and ice caps exhibit a pronounced negative surface mass balance,

contributing to sea level rise. Perhaps equally important are the local scale changes occurring in glaciated catchments where the volume and timing of meltwater affects the local environment both on land and in fjords and oceans. In-situ observations



of surface mass balance processes are important for understanding the effect of future climate change (e.g. Machguth et al., 2013) and while Greenland ice sheet ablation zone is well monitored by the in-situ network of automatic weather stations run by the Programme for Monitoring the Greenland Ice Sheet (PROMICE, Fausto et al., 2021) and the interior of the ice sheet

monitored by the Greenland Climate network (GC-Net, Vandecrux et al., in review), only very few of the peripheral glaciers distinct from the Greenland ice sheet are being monitored. Due to local effects of peripheral glaciers being in coastal areas in complex terrain, there is a strong difference in surface mass balance between peripheral glaciers and the main ice sheet (Abermann et al., 2019) and peripheral glaciers have already passed the tipping point for meltwater retention and runoff that the main ice sheet is yet to experience (Noël et al., 2017). This all sum up to a contribution to sea level rise is from peripheral

glaciers and ice caps that is disproportionately high compared to the area and mass of these glaciers in relation to the main ice sheet (Bolch et al., 2013; Hugonnet et al., 2021).

The data presented here is from a transect of three Automatic Ablation and Weather Stations (AAWSs) located on the A. P. Olsen ice cap (referred to here as APO or the Ice Cap), NE Greenland (Figure 1). The transect is part of the GlacioBasis Zackenberg glaciological monitoring programme, which is a subprogram of the Greenland Ecosystem Monitoring (GEM,

g-e-m.dk) at Zackenberg Research station, located in the Northeast Greenland National park. The Greenland Ecosystem Monitoring (GEM) is an integrated monitoring and long-term research programme on ecosystems and climate change effects and feedback mechanisms in the Arctic. GEM covers three sites representing three zones of the Greenland arctic area: Zackenberg in Northeast Greenland (High arctic), Disko island in Central west Greenland (transition zone between high arctic and low arctic) and Nuuk in Southwest Greenland. The Zackenberg site is the longest running site where ecosystem monitoring has

been ongoing since 1995, and GlacioBasis Zackenberg is the longest running glaciological monitoring program in GEM. APO was chosen for glaciological monitoring because it is the largest contributor of glacial meltwater into the Zackenberg River, which plays a crucial role in the downstream ecosystem, including the Young Sound ecology (Citterio et al., 2017; Sejr et al., 2022).

The first two AAWSs of the APO transect were installed in late April 2008 in the ablation zone, whereas the third AAWS

was installed in August 2009 in the accumulation zone at the Ice Cap summit. These AAWSs have been running with alternating instrumentation until April 2022. In spring 2022 installation of new standardized AAWSs was initiated, these stations are similar to the PROMICE and GC-Net stations (Fausto et al., 2021). With the new standardized setup, the data from the APO transect will be handled as a PROMICE and GC-Net dataset and data processing will be done using the python package pypromice described in How et al. (2023). The purpose of this paper is to describe the dataset collected from the APO transect

in the period before the standardized setup: May 2008 through May 2022. The variables published here are: Ice ablation, air temperature, relative humidity, air pressure, wind speed, incoming and outgoing shortwave and longwave radiation as well as AAWS tilt, snow depth and the derived variables cloud cover fraction, surface temperature and albedo. These variables capture the major components of the surface energy balance, and thus the data can be used to study processes governing surface mass balance. Additionally, this dataset can be used to force and calibrate distributed surface ice ablation models such as the

Distributed Surface Energy Balance Model (Hock and Holmgren, 2005) or COSIPY (Sauter et al., 2020). Furthermore, the variables are considered essential climate variables by the the World Meteorological Organization's Global Climate Observing



System (GCOS). Most importantly in-situ observations of near surface climate and ablation are available from very few peripheral glaciers distinct from the Greenland ice sheet in Greenland, and a transect of three AAWSs is, to the current knowledge of the authors, unique to Greenland. The APO transect contributes to the network of Automatic climate observations done by
GEM in the Zackenberg Valley, and can be used in studies combining data from different surfaces such as the the study of temperature slope lapse rates in Shahi et al. (2023) and the spatiotemporal variability in surface energy balance in Lund et al. (2017).

The paper is organized as follows: Section 2 provides an overview of the study area, including logistical conditions for field visits. Section 3 describes the details behind the collection of data and the post-processing done. Section Section 4 describes the
quality control and data filtering. Section 5 demonstrates the suitability of these data for energy balance calculations. Sections 6 are concluding remarks summarizing the paper. These sections are followed by information about processing scripts and data availability.

## 2 Study area and monitoring setup

APO is an ice cap with several glacier catchments extending in elevation from around 200 to 1500 m a.s.l., and covering
a total area of about 300 $km^2$. The glacier catchment in this paper labeled East in Figure 1, for reference the Randolph Glacier Inventory (RGI) ID is: RGI60-05.20098 (RGI Consortium, 2017), is the main contributor of glacial meltwater in the Zackenberg River catchment, and thus the area of focus for the glaciological monitoring(Figure 1). The APO transect consists of three AAWS sites (see Figure 1 and Table 1): the lower site, $ZAC\_L$ where L refers to the lower ablation zone, has the longest and the fullest data record. The middle site, $ZAC\_U$ where U refers to the upper ablation zone, is located as close
to the equilibrium line altitude as logistically possible. $ZAC\_U$ initially had a limited number of instruments. The top site, $ZAC\_A$ where A refers to the accumulation zone, is located at the Ice Cap summit at an elevation of 1477 m. The COVID-19 pandemic travel restrictions in 2020 and 2021 resulted in the burial of AAWS at $ZAC\_A$ in 2020 and the station has yet not been recovered.

Due to the remote location, the ice cap can mainly be reached by snow scooters traveling from Zackenberg Research station,
limiting the period where the glacier can be visited to the short period in spring after sunrise and before snow melt, usually the last two weeks of April. This means that the maintenance of the AAWSs is sensitive to snow conditions in April, and with the limited access data gaps are inevitable. Please note that ice cap surface mass balance from the AAWSs and a transect of stakes in the ablation zone is reported to the World Glacier Monitoring Service (wgms.ch) every year. Due to a discrepancy in the definitions of glacier catchments, $ZAC\_L$ and $ZAC\_U$ is in the East catchment (RGI ID: RGI60-05.20098) but $ZAC\_A$ is
attributed to the RGI ID: RGI60-05.20092, labeled the North catchment in Figure 1.



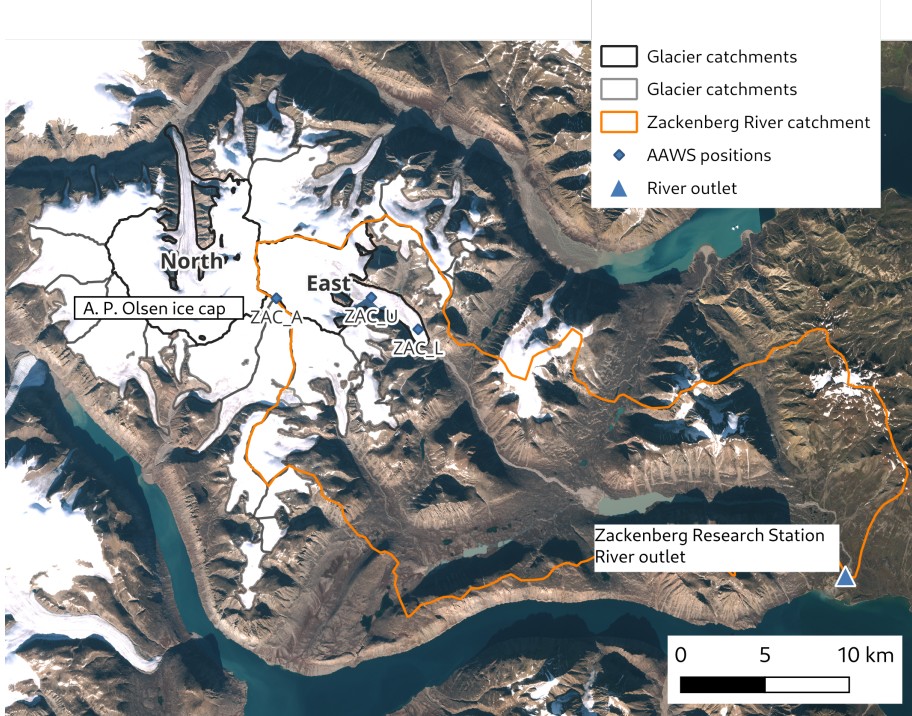

**Figure 1.** A. P. Olsen ice cap outlined in individual glacier catchments modified slightly but following the Randolph Glacier Inventory (RGI Consortium, 2017) and the hydrological catchment of Zackenberg River (orange outline). The base map is from the European Space Agency (ESA) Sentinel 2 satellite in 2022; the AAWS are marked with blue diamonds; and the hydrometric station close to the river outlet is marked by a blue triangle.

**Table 1.** Elevation, position and monitoring start date of the three AAWSs on the A. P. Olsen transect.

| Station | Elevation | Latitude | Longitude | Start year |
|---------|-----------|----------|-----------|------------|
| $ZAC\_L$ | 694 m a.s.l. | 74.6241 N | 21.3742 W | 2008 |
| $ZAC\_U$ | 920 m a.s.l. | 74.6434 N | 21.4619 W | 2008 |
| $ZAC\_A$ | 1477 m a.s.l. | 74.6475 N | 21.6520 W | 2009 |

## 3 Instruments and methodology

In this section we describe the instrumentation on the AAWS as well as the steps taken to go from raw observations to filtered and quality checked data. Table 2 provides an overview of variables and the names used both in the text and in the data files. Variable names in the data files match the names used in PROMICE/GC-Net (How et al., 2023).





**Table 2.** Variables and their respective names and units in this paper and the data files. Naming convention in the data files follow the names given in the PROMICE/GC-Net data.

| Observed variables | Name in this paper | Name in csv file | Unit |
|---|---|---|---|
| Air temperature | $T_{air}$ | $t\_u$ | $^{\circ}C$ |
| Relative humidity | $RH_{corr}$ | $rh\_corr$ | $\%$ |
| Air pressure | $P_{air}$ | $p\_u$ | $hPa$ |
| Shortwave incoming radiation | $SR_{in}, SR_{in\_corr}$ | $dsr, dsr\_corr$ | $Wm^{-2}$ |
| Shortwave outgoing radiation | $SR_{out}, SR_{out\_corr}$ | $usr, usr\_corr$ | $Wm^{-2}$ |
| Longwave incoming radiation | $LR_{in}$ | $dlr$ | $W\ m^{-2}$ |
| Longwave outgoing radiation | $LR_{out}$ | $ulr$ | $W\ m^{-2}$ |
| Wind speed | $WS$ | $wspd$ | $m\ s^{-1}$ |
| Surface height (snow depth) | $Z_{boom}$ | $z\_boom$ | m |
| Ice ablation, pressure transducer assembly | $Z_{pta}$ | $ice\_ablation$ | m ice |
| Ice ablation, sonic ranger | $Z_{stake}$ | not included | m ice |
| Station tilt | $Tilt_x, Tilt_y$ | $tilt\_x, tilt\_y$ | degree |
| Derived variables | | | |
| Albedo | $\alpha$ | $albedo$ | unitless |
| Cloud cover fraction | $cloud\_cover$ | $cloud\_cover$ | $\%$ |
| Surface temperature | $T_{surf}$ | $t\_surf$ | $^{\circ}C$ |
| Irradiance (top of atmosphere) | $I_{toa}$ | $I$ | $W\ m^{-2}$ |

## 3.1 Automatic ablation and weather station design

The AAWSs are designed as free floating tripods (Figure 2, left) and the height of the instruments is reduced when snow accumulates during winter (Figure 2, right). In the ablation zone the snow melts away completely every summer and thus the distance to the surface annually reaches it's maximum value. In the accumulation zone the instruments are lifted manually during field visits and the distance to the surface is more variable.

To conserve power the data logger the AAWS is dormant and power up at 10 min intervals, where instantaneous values for all variables are collected. The only exception to this is wind speed since winds speed is measured by the number of rotations of the propeller since last data collection and thus the wind speed observation represents an average over the past 10 min. The data is published as hourly averages for hours where all six instantaneous observations are available.





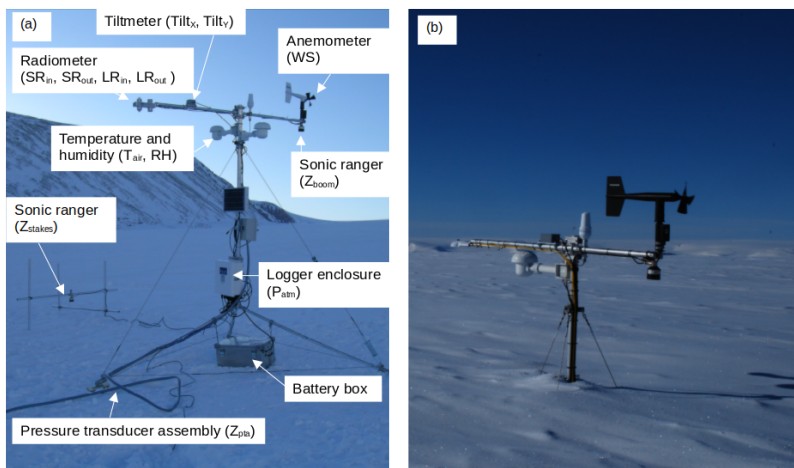

**Figure 2.** Panel (a): Photo of $ZAC\_L$ from installation in 2008 with labels showing the location of the instruments collecting the key variables published here. Panel (b): Photo of $ZAC\_A$ from the field visit in April 2012, illustrating the gradual decrease of sensor height due to snow accumulation. Photo credit: Michele Citterio

**Table 3.** Instrument types, uncertainty and average maintenance schedule for the instruments installed at the three AAWSs on the A. P. Olsen transect.

| Instrument type | Manufacturer | Model | Accuracy | Maintenance |
|---|---|---|---|---|
| Barometer | Campbell Scientific | CS100/Setra 278 | $\pm 2hPa$ | 5 years |
| Thermometer, aspirated | Rotronic in rotronic assembly | MP100H-4-1-03-00-10DIN | $\pm 0.1K$ | 5 years |
| Hygrometer | Rotronic in rotronic assembly | Hygro Clip HC2 | $\pm 0.8\%$ | 1-2 years |
| Anemometer | R. M. Young | 05103-5 | $\pm 0.2ms^{-1}$ or 1% of reading | 3 years |
| Radiometer | Kipp and Zonen | CNR1 or CNR4 | $\pm 10\%$ | 4 years |
| Sonic ranger | Campbell Scientific | SR50A | $\pm 1cm$ or $0.6 - 0.8\%$ | 1-2 years |
| Pressure transducer | Ørum & Jensen in GEUS assembly | NT1400 | $\pm 2.5cm$ | 5 years |
| Inclinometer | HL Planar in GEUS assembly | NS-25/E2 | $0.6\%$ | 5 years |

## 3.2 Temperature and humidity

Air temperature ($T_{air}$) and relative humidity are measured in a radiation shield equipped with a fan for forced ventilation. The instrument is placed at a height approximately 2.6 m above the tripod feet. Temperature is measured with a PT100 and relative humidity with Rotronic HygroClip. The HygroClip is replaced at each field visit with an instrument re-calibrated in a closed chamber at room temperature with constant relative humidities of 10%, 35% and 80%. The relative humidity is measured





relative to the maximum saturation of air thus, relative to liquid water which is valid at temperatures above freezing. For temperatures below the freezing point, the observed relative humidity ($RH_{obs}$) is recalculated relative to ice using the method described in Goff and Gratch (1946):

$$RH_{corr}(T_{air} < 0) = RH_{obs}(T_{air} < 0)\frac{es_{water}}{es_{ice}} \tag{1}$$

where $es_{ice/water}$ is the saturation water vapor pressure over ice or water. Relative humidity is filtered to contain only values between 0 and 100%.

The hourly average of relative humidity is calculated from averaging the vapor pressure ($e$) and then calculating back to relative humidity. The relation between vapor pressure and relative humidity is given by (based on Lowe, 1976):

$$RH = 100 * \frac{e}{e_s} \tag{2}$$

where, $RH$ is relative humidity and $e_s$ is specific humidity relates to air temperature $T$ via

$$e_s = \alpha_0 + \alpha_1 T + \alpha_2 T^2 + \alpha_3 T^4 + \alpha_4 T^4 + \alpha_5 T^5 + \alpha_6 T^6 \tag{3}$$

See the given values for $\alpha_0$ to $\alpha_6$ in Appendix A.

### 3.3 Radiation and derived variables

The four radiation components, incoming and outgoing, short and long wave radiation are observed using a Kipp and Zonen CNR1 and CNR4 installed approximately 2.6 m above the tripod feet. Instruments are replaced with newly calibrated instruments every 4 years, when logistically possible. The AWS tripod is floating freely on the ice surface in the ablation zone and both tilt and direction vary as the surface melts. The movement of the station affects in particular the recorded incoming and outgoing shortwave radiation and thus the radiometer is accompanied by a tilt meter making us able to correct for the tilt of the instrument.

#### 3.3.1 Correction incoming shortwave radiation for tilt and deriving cloud cover

The tilt correction of incoming solar radiation follows van As (2011). Incoming shortwave radiation ($SR_{in}$) can be split into a diffuse fraction ($f_{diff}$) and a direct fraction. The diffuse radiation is not affected by the tilt of the instrument and so it is only the direct beam part that is corrected:

$$SR_{in\_corr} = SR_{in}\frac{C}{1 - f_{diff} + Cf_{diff}} \tag{4}$$



$$
\begin{aligned}
C = \cos(SZA)(&\sin(d)\sin(lat)\cos(\phi_{sensor}) \\
&- \sin(d)\cos(lat)\sin(\theta_{sensor})\cos(\phi_{sensor}+\pi) \\
&+ \cos(d)\cos(lat)\cos(\theta_{sensor})\cos(w) \\
&+ \cos(d)\sin(lat)\sin(\theta_{sensor})\cos(\phi_{sensor}+\pi)\cos(w) \\
&+ \cos(d)\sin(\theta_{sensor})sin(\phi_{sensor}+\pi)\sin(w))^{-1}
\end{aligned}
\tag{5}
$$

where $SZA$ is the solar zenith angle, $d$ is the sun declination, $w$ is the hour angle (see procedures for calculating $SZA$, $d$ and $w$ in Vignola (2019)), $lat$ is the instrument latitude in radians and $\phi_{sensor}$ and $\theta_{sensor}$ are the tilt angle and direction, respectively.

The tilt corrected values are passed through a filter removing spikes exceeding top of atmosphere irradiance given by:

$$
I_{toa} = I_0 \cos(SZA)
\tag{6}
$$

Where $I_0 = 1361 W m^{-2}$ is the solar constant.

The diffuse fraction of the incoming shortwave radiation ($f_{diff}$) ranges from 0.2 to 1 corresponding to clear skies and fully overcast conditions, respectively, and we assume a linear relationship to the cloud cover fraction ($Cloud\_cover$).

The cloud cover fraction is calculated based on its dependence on air temperature ($T_{air}$) similar to the approach of van As et al. (2005). Firstly, the theoretical clear sky incoming longwave radiation, $LR_{clear}$, is calculated based on Swinbank (1963):

$$
LR_{clear} = 5.31 \cdot 10^{-14}(T_{air} + T_0)^6
\tag{7}
$$

where $T_0 = 273.15°C$. Secondly, for theoretical overcast conditions, $LR_{overcast}$, black body radiation is assumed:

$$
LR_{overcast} = 5.67 \cdot 10^{-8}(T_air - T_0)
\tag{8}
$$

The cloud cover fraction is thus:

$$
Cloud\_cover = \frac{LR_{in} - LR_{clear}}{LR_{overcast} - LR_{clear}} = \frac{f_{diff}^{-0.2}}{0.8}
\tag{9}
$$

And hence:

$$
f_{diff} = 0.2 + 0.8 \cdot Cloud\_cover
\tag{10}
$$

The radiometer is repositioned towards south at every field visit. However, during the melt period the station can change azimuth direction and the exact direction of the instrument is not measured beyond the yearly field visits, which causes an uncertainty that is not quantified. This is addressed in the quality control in a later section.





### 3.3.2 Deriving albedo

The albedo is given by

$$albedo = SR_{out}/SR_{in} \qquad (11)$$

and filtered to include only data when the sun is in view of the upper sensor, which is when the angle between the sun and the sensor ($AngleDif$) is below $70°$ and $SZA$ above $70°$. $AngleDif$ is given by:

$$AngleDif = 180/\pi \arccos(\sin(SZA)\cos(w+\pi)\sin(\theta_{sensor})\cos(\phi_{sensor})$$
$$+ \sin(SZA)\sin(w+\pi) * \sin(\theta_{sensor}) * \sin(\phi_{sensor}) + cos(SZA) * cos(\theta_{sensor})) \qquad (12)$$

The gaps in the albedo record are filled using a forward fill function in order to use the albedo to correct the outgoing shortwave radiation as described below.

### 3.3.3 Correcting outgoing shortwave radiation

The radiation sensor has limitations when the sun angle is low and the sun beams hit the lower sensor intended to record outgoing shortwave radiation. When the sun is in the field of view of the outgoing sensor, it is assumed that the incoming

sensor only records diffuse radiation. It is assumed that the sun is in view of the outgoing sensor when $AngleDif$ below $90°$ and SZA above $90°$. The outgoing shortwave radiation is in this case calculated using the albedo:

$$SR_{out} = \frac{albedo}{f_{diff}}, \text{ if } AngleDif \ < \ 90° \text{ and } SZA \ > \ 90° \qquad (13)$$

### 3.4 Air pressure

The barometer is a Campbell Scientific CS100/Setra 2078 placed inside the fiberglass-reinforced polyester logger enclosure

located around $1.5\ m$ from the ice surface. A porous vent filter equalizes pressure inside and outside the logger enclosure. The measurement uncertainty of the instrument is reported to be $2\ hPa$ in the range of -40 to $60°C$.

### 3.5 Wind speed

The anemometer is the model 05103-5 from R. M. Young. The instrument is placed approximately 3 m above the tripod feet. The accuracy of the instrument is $0.3\ ms^{-1}$ up to wind speeds of $30\ ms^{-1}$, above the accuracy is $1\%$.

### 3.6 Snow depth/sensor height

The distance between the surface the instruments and effectively the snow height is measured using a sonic ranger manufactured by Campbell Scientific (model SR50A) mounted on the AAWS boom. The sonic ranger detects the distance to the surface by recording the travel time of reflected sonic waves. The sonic wave speed in air depends on air temperature and thus the observed





distances ($Z_{boom\_raw}$) are corrected for air temperature ($T_{air}$):

$$Z_{boom} = Z_{boom\_raw} \sqrt{\frac{T_{air} + T_0}{T_0}},$$ (14)

where $T_0 = 273.15°C$. The accuracy of the instrument was found by Fausto and van As (2012) to be between 0.6 - 0.7 % using observations from the Greenland Ice Sheet. The manufacturer reports an uncertainty of 0.4 %.

## 3.7 Ice ablation

Ice ablation is observed continuously, mainly using the pressure transducer assembly (PTA) described in (Fausto and van As, 2012). A pressure transducer is installed at the end of a hose and the hose is filled with antifreeze liquid. The hose is drilled into the ice at a usual depth of 10 to 14 $m$. When the ice melts the surface lowers and the pressure drop in the liquid column of the hose is converted and can be converted into a surface lowering. The pressure transducer signal, $Z_{pta}$, is affected by air pressure and corrected using the following equation:

$$Z_{pta\_corr} = Z_{pta} \frac{P_C - P_{air}}{g\rho_l},$$ (15)

where $P_C$ is the the calibration pressure provided by the manufacturer in $hPa$, $P_A$ is the air pressure in $hPa$, $g = 9.81ms^{-2}$ is the gravitational constant and $\rho_l$ is the density of the antifreeze liquid in the hose. The uncertainty of the instrument is estimated to be 4 cm and contains some noise. For the purpose of making the data easy to use the ice ablation is set to zero at the beginning of every melt season. This is done by subtracting the mean of a week prior to the onset of ice melt.

Supplementary to the pressure transducer assembly a sonic ranger is mounted on separate stakes drilled into the ice. The surface height measured from sonic ranger is corrected for air temperature and is denoted $Z_{stake}$.

## 4 Data quality, uncertainty and filtering

In the subsequent sections, we first detail major station failures, followed by an in-depth discussion on the quality and uncertainties tied to each specific variable. Our quality control process primarily involves a visual inspection of the data to identify outliers and detect data drift. Furthermore, we compare variable gradients across the three AAWSs to pinpoint periods with potentially problematic data. The success rate of our measurements after data filtering is depicted in Figure 3.

Quality control is done to the best of our current knowledge, but the data is considered living data and should be in a state to be used directly for the continued monitoring of the A. P. Olsen transect. This means that corrections and filtering of data might change in future versions of the dataset. The filtered data could offer significant insights, and this is therefore included as supplementary data.

## 4.1 Major station failures

Reviewing the raw data and field notes, it becomes evident that several major events led to data loss across all variables, as described in the following.



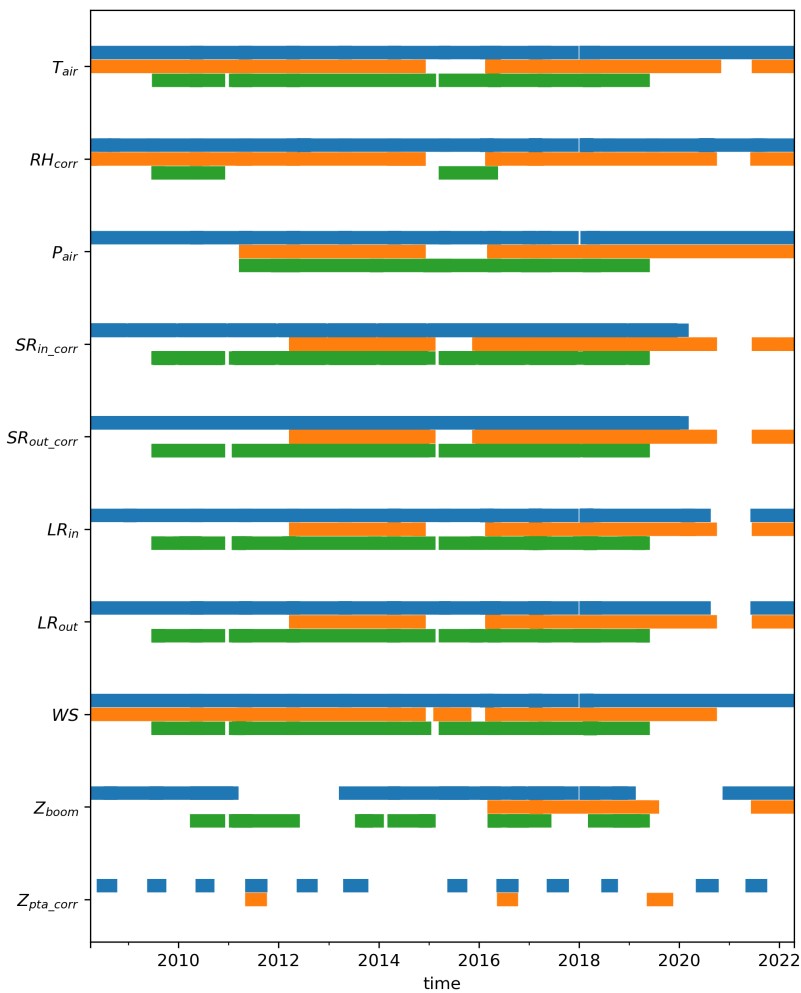

**Figure 3.** Measurement success rate for the 10 key variables, blue is $ZAC\_L$, orange is $ZAC\_U$ and green in $ZAC\_A$

In 2015, $ZAC\_U$ tipped over and was subsequently erected in April 2016. This incident is evident in the dataset as poor quality data, and data from all variables are removed for this period. $ZAC\_U$ tipped over again in 2020 and was erected and underwent repairs in July 2021. Data from this period has also been filtered out. During the winter of 2010/2011, $ZAC\_A$ tilted or got snow covered and part of the data was lost. In January 2015, most instruments at $ZAC\_A$ were buried by snow, only to be excavated in April 2015, these data are also filtered out. The $ZAC\_A$ record ends in April 2019, marking the final





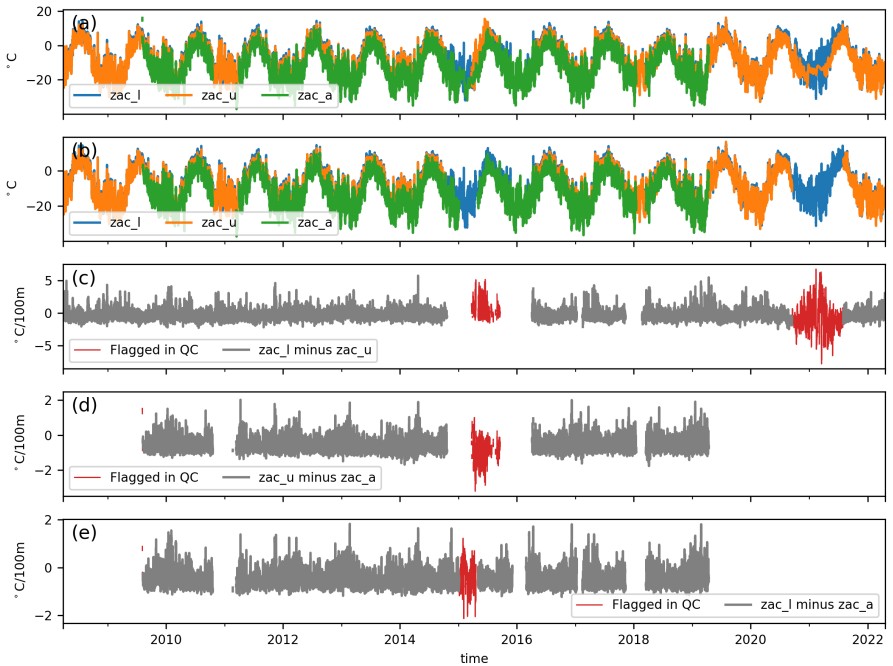

**Figure 4.** Air temperature quality control. Panel (a) and (b): Unfiltered and filtered data respectively, $ZAC\_L$ is blue, $ZAC\_U$ is orange and $ZAC\_A$ is green. Panel (c) to (e): Gradients between $ZAC\_L$ and $ZAC\_U$, $ZAC\_U$ and $ZAC\_A$, $ZAC\_L$ and $ZAC\_A$ respectively.

visit before the station was entirely buried in snow and could not be reached due to travel restrictions imposed during the Covid-19 pandemic.

**4.2 Temperature**

Temperature observations rely on the instrument casing being adequately ventilated. However, the ventilation fan consumes a significant amount of power and is deactivated when battery levels are low. This most often happens during winter when the batteries cannot be re-charges due to the polar night, coinciding with the period where ventilation of the casing is not necessary. Thus, we consider the effect of this to be minor and we have not detected any problems with the data due to this.

To evaluate the data quality of temperature readings, we compare data year-over-year and examine the gradients in values between stations, as depicted in Figure 4. This figure highlights the impact of the instrument burial at $ZAC\_A$ in 2015, which is evident from an unusual negative temperature gradient between $ZAC\_L$ and $ZAC\_A$ (see Figure 4, panel (e)). Additionally, the tilting incidents at $ZAC\_U$ in 2015 and 2020 manifest as unusually high and low lapse rates between $ZAC\_U$ and $ZAC\_L$ and $ZAC\_U$ and $ZAC\_A$ (see Figure 4, panel (c) and (d)). Besides the major station problems we found no quality issues 220 with the air temperature observations.





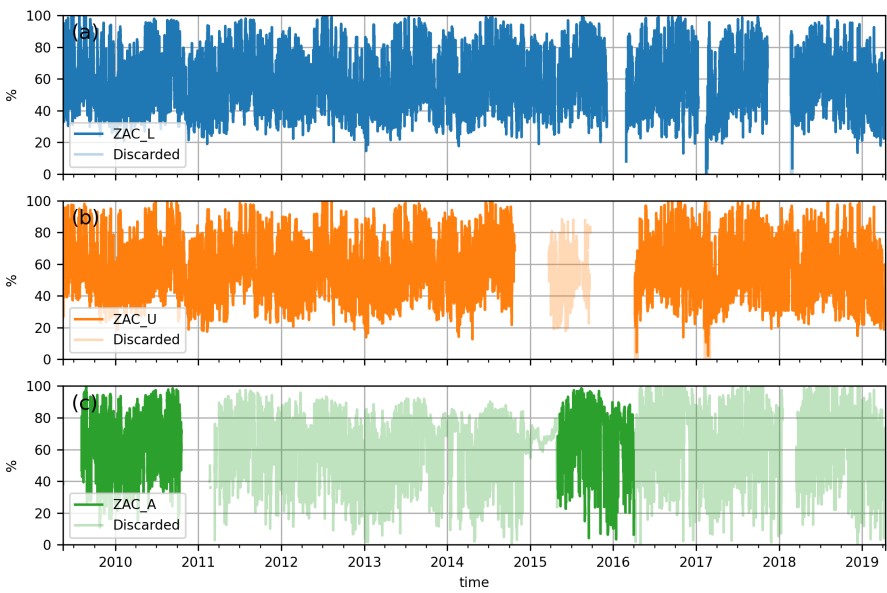

**Figure 5.** Relative humidity ($RH_{corr}$) at $ZAC\_L$ in panel (a), $ZAC\_U$ in panel (b) and $ZAC\_A$ in panel (c)

## 4.3 Relative humidity

The humidity sensor typically requires recalibration every 1-2 years. However, due to logistical challenges, this was not always feasible and an uncalibrated sensor will drift towards increasingly poorer performance. Drifting values of relative humidity is observed at $ZAC\_A$ during 2012-2014 (Figure 5), which we believe is due to an uncalibrated HygroClip. While the HygroClip was replaced in 2015, data from 2016 raised concerns as RH frequently reached 100%, more often than at other stations. The cause of the drift in relative humidity values from 2016 at $ZAC\_A$ is unknown, and the affected data has been discarded.

## 4.4 Shortwave radiation and tilt

The radiation sensor can be affected by shorter periods with riming causing outliers which is partly dealt with by removing outliers beyond fixed thresholds as described for the individual variables. The radiation sensors also face issues related to high tilt and azimuth misalignment from south, we assumed that the effect of this is negligible on the longwave radiation component, but on the shortwave component tilt can have a significant effect. During each field visit, the AAWS is adjusted to ensure the radiometer faces south. However, as the AAWS floats on the surface, it can tilt as well as rotate at varying degrees during the melt season. While the shortwave radiation is corrected for tilt, the correction does not take azimuth misalignment into account. If the sensor turns more towards the west or east, the tilt correction can become inaccurate, as it operates under the assumption that the sensor is oriented southward. The uncertainty of the the tilt-corrected shortwave radiation, can be evaluated by investigating the total tilt, the size of the correction by comparing $SR_{in\_corr}$ with $SR_{in}$ as well as comparing the corrected values to potential incoming radiation as done in the following.


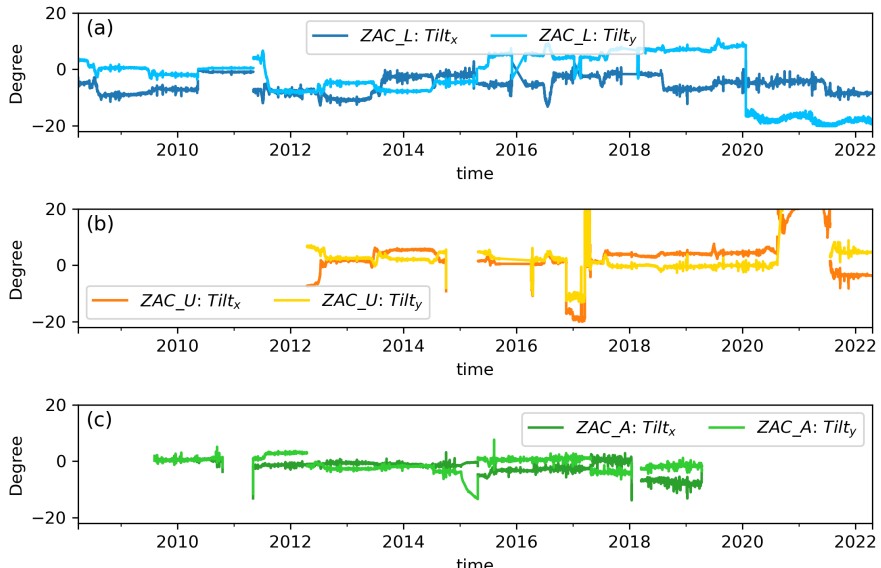

**Figure 6.** Tilt of the AAWS boom at $ZAC\_L$ in panel (a), $ZAC\_U$ in panel (b) and $ZAC\_A$ in panel (c)

Figure 6 displays the x and y components of the measured tilt. Typically, the AAWS tilt does not exceed an absolute value of 10°. Exceptions to this are instances when a station has been entirely tipped over or buried in snow. The tilt is most variable at $ZAC\_L$ which is in line with observations of a very uneven surface upon fields visits. $ZAC\_A$ is more stable due to the fact that it is positioned in the accumulation zone and therefore is stabilized by the snow. In January 2020, a shift in $Tilt_y$ at $ZAC\_L$ occurred, from field notes this can be explained by damage to the tripod legs and following loosening of the guy wires, after snow cover. The non-tilt corrected data could potentially provide information discerning cloud cover variations, but using this should be approached with caution as absolute values are not reliable. The tilt corrected incoming shortwave radiation is shown in Figure 7. The peak values of the data vary a lot at $ZAC\_L$ and while this could be due to natural variations as $ZAC\_L$ is located low in a valley that is prone to have low clouds, this might also be due to poor data quality. Thus, in order to evaluate the success of the tilt-correction and the quality/uncertainty of the radiation data we compare corrected and non-corrected shortwave incoming radiation in Figure 8. The top of atmosphere irradiance ($I_{toa}$, Equation 8) is used as a visual guideline in the comparison. Panel (a) in Figure 8 with data from $ZAC\_L$ in 2009, shows a successful year where the tilt correction modifies the values slightly. Panel (b) in Figure 8 shows a year where the tilt of the station has been more severe and uncertainties must be assumed higher in such years. Specifically at $ZAC\_L$ the years spanning 2012 to 2016 and 2018 to 2020 needed to be corrected more than other years, and uncertainty is expected to be higher for these years. Figure 8 also shows the minimum values of observed $SR_{in}$ are ranging below the minimum $I_{toa}$, indicating a substantial diffuse component.

Finally, the quality of incoming and outgoing shortwave radiation is evaluated by comparison with remotely sensed albedo values. The albedo values used are from the Google Earth Engine Albedo Inspector (https://www.glacier-hub.com/posts/GEE-



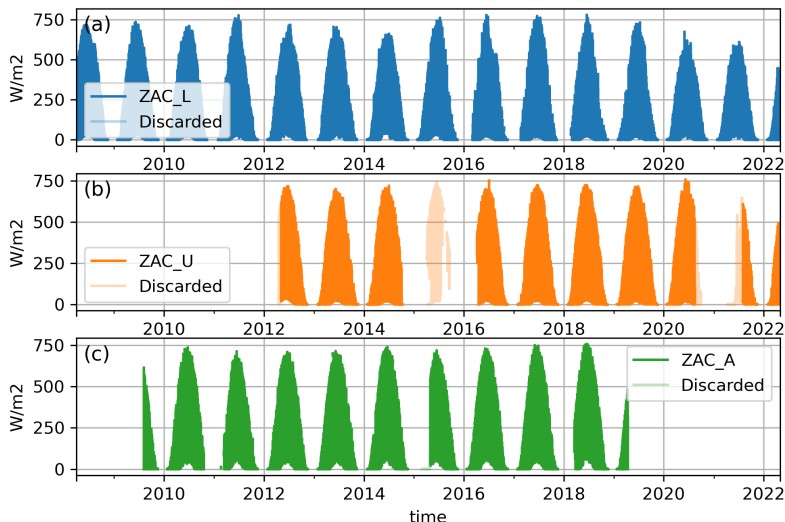

**Figure 7.** Incoming shortwave radiation corrected for tilt ($SR_{in\_corr}$) at $ZAC\_L$ in panel (a), $ZAC\_U$ in panel (b) and $ZAC\_A$ in panel (c).

toolbox-for-glacier/) based on the work done by Feng et al. (2023) in Figure 9. The comparison between a point measurement from the AAWS with a grid value introduces an uncertainty. There is a generally good correlation between the in-situ and remotely sensed albedo values with a goodness of fit, $R = 0.55$, which is comparable to the values obtained by Feng et al. (2023) when comparing the satellite derived albedo with PROMICE data.

### 4.5 Longwave incoming and outgoing radiation

The incoming longwave radiation ($LR_{in}$) is automatically filtered to remove data lower than $120\ Wm^2$ and the outgoing longwave radiation ($LR_{out}$) is filtered to remove data lower than $150\ Wm^2$. The outliers are believed to occur on riming events and choice of limits is based on a visual assessment of outliers (Figure 10).

There is a period between July 2020 to July 2021 at $ZAC\_L$, where the longwave radiation data look substantially higher that the rest of the period. The cause of this remains elusive, and the data is filtered out.

### 4.6 Air pressure and wind speed

We saw no quality issues with air pressure and wind speed data, and only data from the periods where the stations have either tipped over or got buried in snow have been filtered out from the air pressure and wind speed data. The air pressure is dependent on absolute elevation of the stations and the elevation values given in this paper are based on a multi-year average of a single frequency GPS on the AAWS.

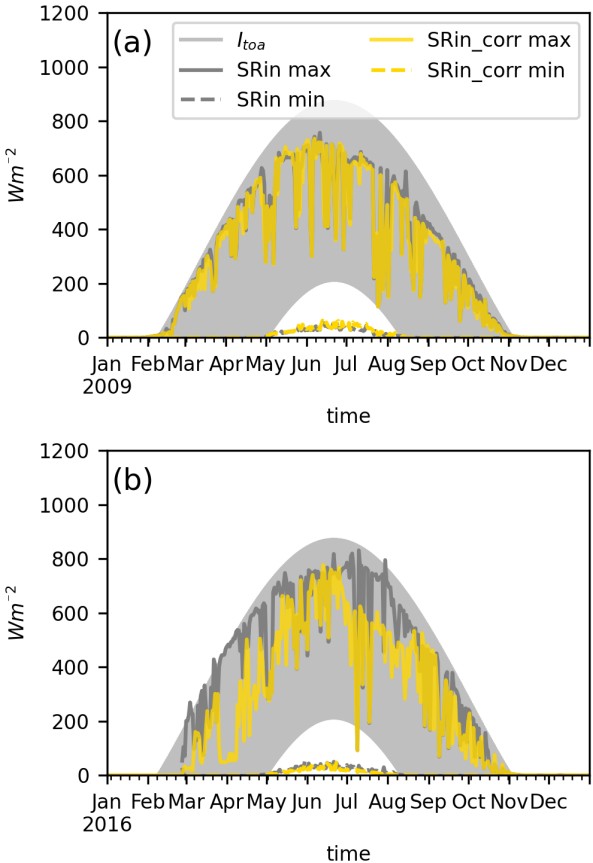

**Figure 8.** $ZAC\_L$: Daily maximum and minimum values of incoming shortwave radiation, corrected and uncorrected for tilt compared with top of atmosphere irradiance. Panel (a) is 2009 and panel (b) is 2016

## 4.7 Ice ablation

The PTA only records ice melt, and the presence of snow cover over the instrument can influence the data. Consequently, all data from October to March is automatically discarded. Instances when the pressure transducer assembly completely melted out of the ice have also been removed, meaning not every year contains a complete melt season. To assess data quality, we compare

the ice ablation observations from the PTA ($Z_{pta}$) with those from the sonic ranger on stakes ($Z_{stake}$). This comparison is limited to $ZAC\_L$ since ice ablation has only been measured by a PTA at $ZAC\_U$, and since $ZAC\_A$ is situated in the accumulation zone.

Overlapping ice ablation data from $ZAC\_L$ spans six years, as shown in Figure 11. In 2008 and 2009, the PTA recorded faster ice ablation rates than the sonic ranger. Notably, in July 2009, the stake assembly holding the sonic ranger collapsed

according to field notes. This incident with a tilting stake assembly might be the cause for the observed lower melt rates by

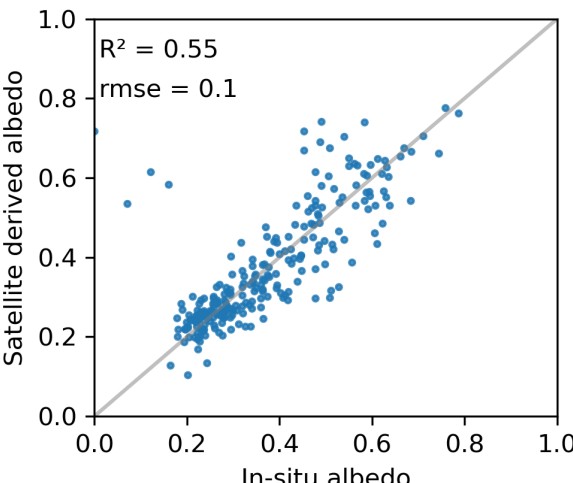

**Figure 9.** Daily albedo values: in-situ observations compared with the satellite derived based on Feng et al. (2023).

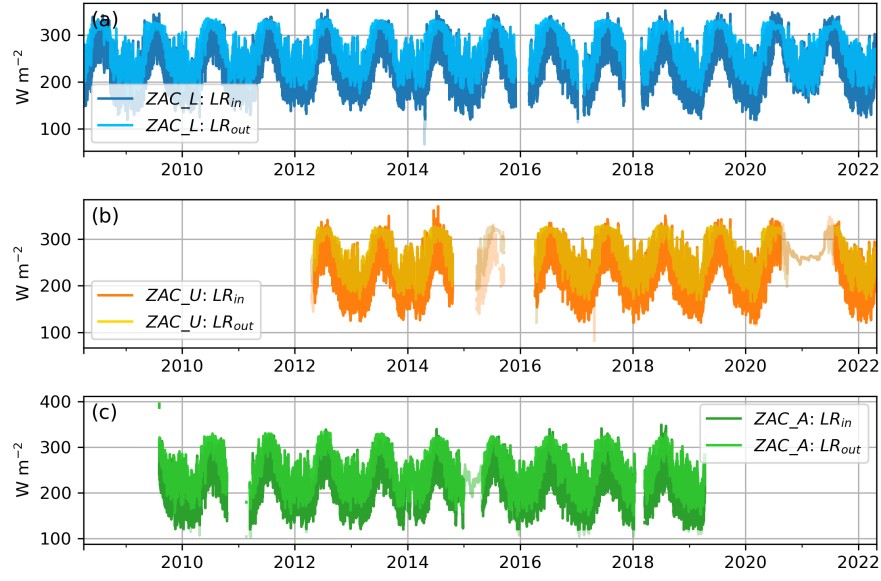

**Figure 10.** Incoming and outgoing longwave radiation ($LR_{in}$, $LR_{out}$) at **a**: $ZAC\_L$, **b**: $ZAC\_U$, **c**:$ZAC\_A$. Pale colors indicate data that has been filtered out.

the sonic ranger. In 2010, the stake assembly was re-established while the PTA setup remained unchanged, and the sonic ranger recorded higher melt rates than the PTA. This indicates no consistent under-catch in the PTA system. In 2012, the melt rates of both systems were similar until late July. The discrepancy might be due to another collapse of the sonic ranger stake





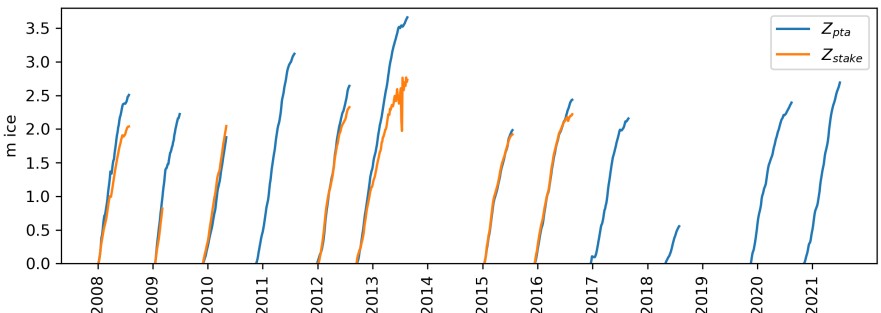

**Figure 11.** Ice ablation recorded using the pressure transducer assembly (PTA, $z_{pta\_corr}$) and the sonic ranger on stakes ($Z_{stake}$) at $ZAC\_L$. Note the a-axis is not continuous, but contains only the months June, July and August for each year and the label refer to July 1 of the given year.

assembly. For 2015 and 2016 the melt rates were closely aligned between the two systems. However, by the end of the 2016

melt season, the two curves diverge. This variation could be due to a snow event visible in the sonic ranger data but not in the PTA. Differences between the two data sets could also arise if they represent distinct surface areas with varying darkness or turbulence conditions.

Generally, we trust the ice ablation from the PTA ($Z_{pta}$) to a higher degree than we trust the sonic ranger observations ($Z_{stake}$), but discrepancies between the two in for example 2012 and 2016 illustrates the uncertainty in the ice ablation obser-

290 vation. Snow events during the ice melt season is not captured by the PTA. This should be kept in mind when using the data for evaluating for example an energy balance model as seen below.

## 5 Use case: A point energy budget melt model

The variables collected at the A. P. Olsen transect are key variables in the surface energy budget equations, and can be used for calculating the energy availability for melting ice. We will exemplify here how a point energy budget melt model is evaluated

with the observed ice ablation. The energy budget model is implemented at $ZAC\_L$ and depends on the observed radiation budget ($SR_{in\_corr}, SR_{out\_corr}, LR_{in}, LR_{out}$), temperature ($T_{air}$), wind speed ($WS$), air pressure ($P_{air}$) and relative humidity ($RH_{corr}$). The use case focuses on two years, 2009 and 2016 where the tilt correction on the radiation data was respectively low and high.

The energy budget is the balance between the net shortwave radiation $SR_{net} = SR_{in} - SR_{out}$, the net longwave radiation

$LR_{net} = LR_{in} - LR_{out}$ and the turbulent heat fluxes: latent heat flux $H_l$ and sensible heat flux $H_s$, as well as the ground heat flux $G$, thus the energy available for melt is given by:

$$Q_{melt} = SR_{net} + LR_{net} + H_l + H_s s + G \tag{16}$$





For the purpose of this example we neglect G and calculate the turbulent heat fluxes following Monin-Obukhov theory (as done in Hock and Noetzli (1997)) where:

$$H_s = c_p \rho_0 \frac{P_{air}}{P_0} \frac{WS \cdot T_{air}}{\ln(z/z_{0w})\ln(z/z_{0t})} \tag{17}$$

and

$$H_l = 0.632 L \kappa^2 \frac{\rho_0}{P_0} \frac{WS \cdot (e_2 - e_0)}{\ln(z/z_{0w})\ln(z/z_{0e})}, \tag{18}$$

where $e_2$ is the vapor pressure at instrument level given by the Clausius–Clapeyron relation:

$$e_2 = 611 \exp\left(\frac{17.27 T_{air}}{243.04 + T_{air}} \frac{RH_{corr}}{100}\right), \tag{19}$$

and $e_0$ is the vapor pressure at a melting surface; $c_p$ is the specific heat of dry air; $L$ is the latent heat of sublimation when $e_2 - e_0$ is negative and the latent heat of evaporation when $e_2 - e_0$ is positive and equal to zero; $\kappa = 0.41$ is von Kármáns constant; $\rho_0$ is the air density at the mean atmospheric level $P\_0$; $z$ is the instrument height here assumed to be constant at 2.7 m; $z_{0w}$, $z_{0t}$, $z_{0e}$ are the roughness lengths for logarithmic profiles of wind, temperature and water vapor, respectively. $z_{0w}$ is kept as a calibration constant and can be varied while $z_{0t}$ and $z_{0e}$ are assumed to be 100 times smaller than $z_{0w}$. All three roughness lengths could be varied to calibrate the model, but this is out of the scope of this example.

The energy surplus is converted to melt by dividing with the latent heat of fusion ($L_f = 334000\ Jkg^{-1}$) so that

$$Melt = Q_{melt}/L_f. \tag{20}$$

This is only valid for a melting surface and sublimation is not accounted for.

The point melt is computed for three values of the surface roughness factor for wind, $z_{0w}$: 0.01, 0.001 and 0.0001, as this value has been shown to vary with orders of magnitude (e.g. Smeets and Broeke, 2008). The performance of the melt model on a daily timescale is evaluated by summing up modeled melt to daily values and taking the difference between the daily mean $Z_{pta}$ value from the day before to the current day to get observed daily values (Figure 12 panel (a) and (b)). The performance of the melt model on a seasonal time scale is evaluated by comparing the accumulated modeled melt with the $Z_{pta}$ with a running daily mean applied to smooth the curve (Figure 12 panel (c) and (d)). The model performance in 2009 shows good agreement with the observed ablation when using $z_{0w} = 0.001$, while the model performs poorly in 2016 for all three roughness factors. This could indicate that the quality of the radiation data is questionable in 2016 and/or that the ice ablation data is spurious.

In conclusion, the presented dataset can be used to evaluate the point melt model for years with high confidence in data, but quality issues, that are not well defined can be limiting the use of the data for such high precision modeling.

## 6 Conclusions

This paper presented the near surface climate and ice ablation dataset from a transect of three automatic weather and ablation stations on the A. P. Olsen Ice Cap in NE Greenland, for the period 2008 through May 2022. The dataset contains all major


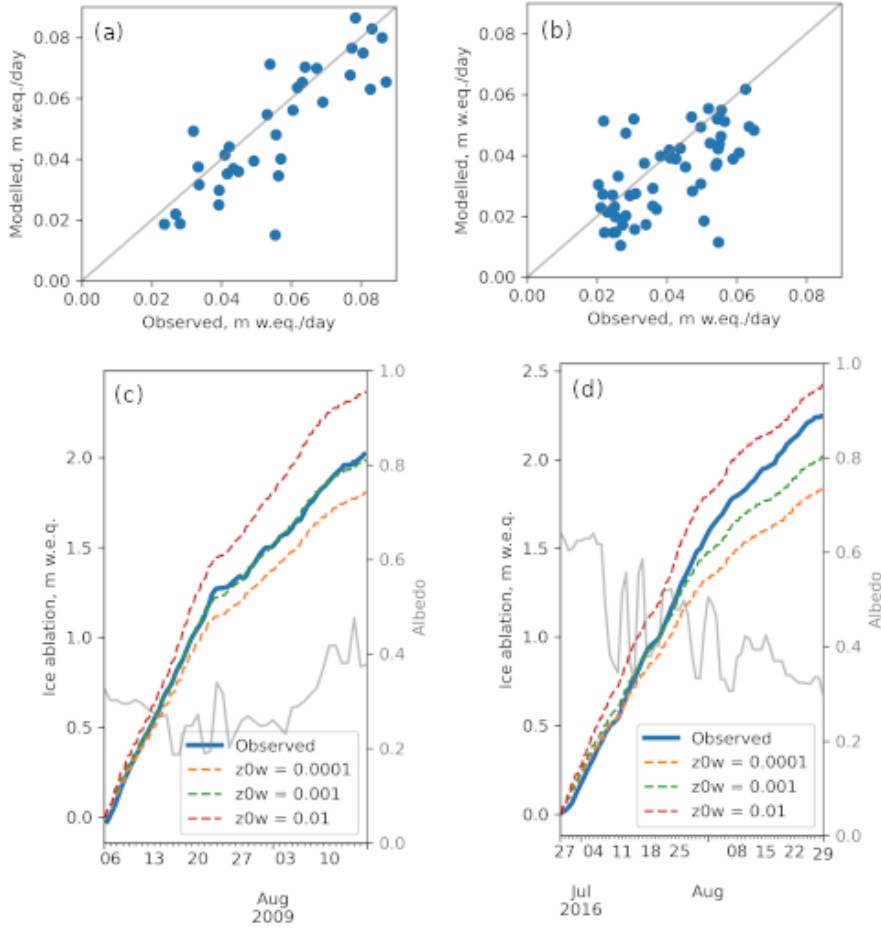

**Figure 12.** Results from the energy budget ice melt model. Panel (a) and (b): Daily modeled ice ablation using the surface roughness for wind: $z_{0w} = 0.001$, compared with observed ice ablation from $Z_{pta}$, for the two years 2009 and 2016 respectively. Panel (c) and (d) show accumulated modeled melt compared with observed ice melt for the three tested values of $z_{0w}$ for 2009 and 2016 respectively.

components of the surface energy balance: Ice ablation, air temperature, relative humidity, air pressure, wind speed, incoming and outgoing longwave radiation as well as the derived variables cloud cover fraction and albedo. The dataset has gone through rigorous instrument corrections and quality control. It can be used to study surface energy budget and ablation processes and to

335 force, calibrate or validate distributed models. Despite the rigorous quality control, uncertainties remain, most importantly for the energy budget calculations are uncertainties in the shortwave radiation and the observed ice ablation. These uncertainties are not quantified in the data set, but exemplified in section 6. The data set is a unique transect of near surface climate on a local ice cap in Greenland and constitutes the first 15 years of a continuous monitoring effort in the Greenland Ecosystem Monitoring programme.





*Code and data availability.* The dataset can be found here: https://doi.org/10.22008/FK2/X9X9GN (Larsen and Citterio, 2023), and in the GEM database: https://data.g-e-m.dk/ (Not updated yet). The data processing code, taking the data from raw to usable data is provided as documentation and can be found at the GitHub site: https://GitHub.com/GEUS-Glaciology-and-Climate/GlacioBasis_AWS_processing. The point energy budget model script can be found here: URL https://GitHub.com/GEUS-Glaciology-and-Climate/GlacioBasis_essd_point_energy_balance_model.

**Appendix A: Appendix A**

The constants used in Equation 3:

$\alpha_0 = 6.107799961$

$\alpha_1 = 4.436518521 * 10^{-1}$

$\alpha_2 = 1.428945805 * 10^{-2}$

$\alpha_3 = 2.650648471 * 10^{-4}$

$\alpha_4 = 3.031240396 * 10^{-6}$

$\alpha_5 = 2.034080948 * 10^{-8}$

$\alpha_6 == 6.136820929 * 10^{-11}$

*Author contributions.* Signe Hillerup Larsen, lead the writing of the manuscript and brought the data from raw measurements into the published format, in some parts utilizing the open source code from the PROMICE workflow. Michele Citterio has designed the monitoring
program as project manager from 2007 to 2021 and collected most of the data with great help from the other co-authors. Daniel Binder, Bernhard Hynek and Anja Rutishauser have contributed with the collection and correction of data. Robert Fausto has helped with utilizing the knowledge from the PROMICE data workflow. All co-authors have contributed to the writing the manuscript.

*Competing interests.* There are no competing interests.

*Acknowledgements.* This work was supported by the Greenland Ecosystem Monitoring programme (g-e-m.dk) via the subprogram GlacioBa-
355 sis, funded by the Danish Environmental Protection Agency and the Danish Energy Agency via Climate and environmental support. GlacioBasis would not exist if it was not for Andreas Peter Ahlstrøms initiative with the first application. We thank Zackenberg Research Station for support. Many people have helped conducting the field work in particular we thank the GeoBasis sub-programme for support in the field as well as field participants through the years: Horst Machguth, Ylva Sjöberg, Morten Langer Andersen, Cristina Gerli and Marek Stibal. The design and maintenance of the A. P. Olsen transect is not possible without the support from the GlacioLab at GEUS, where the
360 design, development and building of certain instruments take place in close collaboration with PROMICE and GC-Net. Some errors in the





raw data was discovered as part of the collaboration with Sonika Shahi at University of Graz, who meticulously went through the data during the quality control process.





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
