# Peer review of "Climate and ablation observations from automatic ablation and weather stations at A. P. Olsen Ice Cap transect, NE Greenland, May 2008 through May 2022"

_Earth System Science Data, 2023_

## Author Comment (AC1)

Dear Editor and reviewers,

Below you find answers to each reviewer separately. We agree to all suggested corrections and found them very helpful. We believe that the manuscript has significantly improved by implementing the suggestions. The major changes are that the revised manuscript does no longer use the melt model to evaluate on data quality, but now only showcase how the data can be used. Secondly, the structure of section 3 has been revised now dividing the instrument details and post-processing.

Please find answers to all reviewer comments below, with the following color-codes:

Blue indicates comments from the authors

Red indicates the original text that has been revised

Green indicates the revised text

With best regards,

Signe Hillerup Larsen

**Referee #1**

Review of: „**Climate and ablation observations from automatic ablation and weather stations at A. P. Olsen Ice Cap transect, NE Greenland, May 2008 through May 2022 "** by Signe Hillerup Larsen, Daniel Binder, Anja Rutishauser, Bernhard Hynek, Robert Fausto and Michele Citterio

**Summary and major points**

The manuscript by Hillerup-Larsen end co-authors presents a 14-year meteorological record from a transect of three automatic weather stations on the A.P. Olsen Ice Cap, peripheral to the Greenland Ice Sheet and located on Greenland's north-east coast. The authors present the 14-year record and data post-processing because AWS design and data processing has recently been changed, necessitating to thoroughly rework the previously collected data.

The authors present a very valuable and unique dataset. The efforts of maintaining the remote installations over such a long time period, under challenging conditions, and making data publicly available, are highly appreciated. It is excellent that the authors post-process the data and clearly demonstrate strength and limitations of the data. While the manuscript addresses all relevant points for a data paper, I do have two major concerns.

Firstly, I suggest revising the structure in Section 3, to a lesser degree in Sections 2 and 4. I suggest structuring Section 3 by describing sensors and the field tasks of all variables first as subsections under Section 3.1 (Subsection 3.2 would become 3.1.1

"Temperature ...", followed by 3.1.2 Radiation measurements, and so on). Any data processing would then be placed in a Section 3.2, which could be entitled "Data Processing" or similar. Currently, the text jumps forth and back between basic field tasks and post-processing. Information on data availability and submission (e,g. to the WGMS) is found at various locations throughout the text. I suggest to first describe the complete data, including post processing and derived parameters, then to state what was done with which data product.

Secondly, I am not convinced by the use case. While I believe that the example is valid, I disagree with the conclusions. The use case shows that measured and modelled melt agree within the bounds of uncertainty, for both years. Likely this would become even clearer if a more complete uncertainty assessment of model input parameters and measured ablation would be done. I am not asking for an uncertainty analysis, but interpretation of the use case needs to be modified. The use case also needs a more thorough discussion.

Thank you for a detailed and constructive review, it is highly appreciated! We agree to all comments and correcting them improves the manuscript significantly. We have restructured section 3 according to recommendations as well as reformulated the use case and what the purpose of this is in both abstract and the use case section. Please see the specific comments in the answer to line by line comments.

**Detailed remarks**

Line 12: Not sure, but abstract is typically written in present tense?

In order to ensure comparable data quality from the old and new monitoring station setups, it was necessary to re-evaluate the data collected between 2008 and 2022

In order to ensure comparable data quality from the old and new monitoring station setups, it is necessary to re-evaluate the data collected between 2008 and 2022

We showed that the inherent uncertainties of the data resulted in an accurate reproduction of ice ablation for just one of the two years

We show that the inherent uncertainties of the data result in an accurate reproduction of ice ablation for just one of the two years

Lines 15 -17: Consider mentioning only one link in the abstract? The two links are slightly confusing.

The final sentence (pasted in below) is moved down to the data and code availability section.

Future refinements will be uploaded as new versions and the continuation of the transect time series are available via https://doi.org/10.22008/FK2/IW73UU (How et al., 2022)

Line 32: "Data" in plural, in particular as you present various parameters?

The data presented here is from a transect

The data presented here are from a transect

Lines 57-62: This appears, at least partially, to be a repetition of statements made earlier, consider merging?

Most importantly in-situ observations of near surface climate and ablation are available from very few peripheral glaciers distinct from the Greenland ice sheet in Greenland, and a transect of three AAWSs is, to the current knowledge of the authors, unique to Greenland. The APO transect contributes to the network of Automatic climate observations done by GEM in the Zackenberg Valley, and can be used in studies combining data from different surfaces such as the the study of temperature slope lapse rates in Shahi et al. (2023) and the spatiotemporal variability in surface energy balance in Lund et al. (2017).

Most importantly in-situ observations of near surface climate and ablation are available from very few peripheral glaciers distinct from the Greenland ice sheet in Greenland, and a transect of three AAWSs is, to the current knowledge of the authors, unique to Greenland. The data from the APO transect has provided valuable insights in combination with on-land climate observations done in the Zackenberg Valley to study temperature slope lapse rates in Shahi et al. (2023) and the spatiotemporal variability in surface energy balance in on different surface types in Lund et al. (2017).

Line 70: Units not in italic.

Corrected

Line 74: "fullest data record": unclear, do you mean the most complete record? Or the record with most parameters?

and the fullest data record

and the most complete

Line 79: "Zackenberg Research station", if a name then capitalize all three words

Zackenberg Research station

Zackenberg Research Station

Line 80: "sunrise", replace with "end of the polar night"

after sunrise

the end of polar night

Lines 82-85: Consider moving to a proposed section that details where and how the data are made available (see major comments)

Line 90: As mentioned in the major comments, the structure needs revision. In the current structure this title mentions something (automatic ablation) that does not appear in the text of 3.1 (it appears then in another section).

Line 92-93: Unclear, do the authors mean that during the melt season the distance to the surface is at its maximum (approx. the height of sensors above the feet of the tripod)? Please rephrase.

The AAWSs are designed as free floating tripods (Figure 2, left) and the height of the instruments is reduced when snow accumulates during winter (Figure 2, right). In the ablation zone the snow melts away completely every summer and thus the distance to the surface annually reaches it's maximum value. In the accumulation zone the instruments are lifted manually during field visits and the distance to the surface is more variable.

The AAWSs are designed as free floating tripods (Figure 2, left) with instruments (see Table 3 for a comprehensive list) mounted on a top boom as well as on the mast. The height of the instruments above the surface is reduced when snow accumulates during winter (Figure 2, right). During the main melt season in the ablation zone, the sensors height above the surface reach maximum as soon as the the snow has melted away and thus in this period the sensor height above the surface is equivalent to the sensor height above tripod feet. In the accumulation zone, where snow does not melt away completely every year, and the instruments are lifted manually during field visits, the distance to the surface is variable throughout the year.

Lines 97-98, structure: This statement ("The data is published...") does not fit with the title of the subsection. Consider adding such a statement to where you describe which data are published where and how.

The sentence has been reformulated and placed in the new section 3.2:

After applying all corrections, hourly averages are calculated for all hours where all six instantaneous observations are available.

Figure 2: Poor graphical quality of both photos. Both are underexposed, (b) is strongly underexposed. This can be corrected in image adjustment software.

I found a better photo for the left hand picture and adjusted the lighting in the right hand picture. The resolution is furthermore slightly improved.

Table 3: This table appears not to be referenced in the text. Please check all other items too.

A reference has been added in the beginning of section 3.1.

Section 3.2 and following: I suggest describing the sensors and the field tasks related to all variables first as subsections under Section 3.1 (3.2 would become 3.1.1 "Temperature ...", followed by 3.1.2 Radiation measurements, and so on). Any data

processing would then be placed in a Section 3.2, which could be entitled "Data Processing" or similar. Data submission and publication could go into a section 3.3. I consider it important that the reader first gets a complete overview which products have been generated before it is explained where these are submitted to / published.

Lines 118/119: Sensor replacement: is this the case for other sensors as well? Table 3 suggests yes, but this appears to be mentioned only here.

A sentence with information of replacement schedule has been added to all variables in the new section 3.1.

Table 3 was actually a bit misleading as the replacement schedule came from a report from the first year and contains also the whished for the continued monitoring. The replacement schedule in Table 3 has thus been edited to match what in fact has happened during 2008 through 2021.

Line 121: "making us able" -> "enabling us"

making us able to correct for the tilt

enabling us able to correct for the tilt

Line 166: All sensors are subject to measuring uncertainty. Why is this mentioned here and not in the text for some of the other sensors? Table 3 provides measuring accuracy for all sensors.

As with the replacement schedule the measuring uncertainty is now given for all variables in the new section 3.1.

Line 168: Height of installation: Information is given for this sensor but not for the others?

As with the replacement schedule the height above tripod feet is now given for all variables in the new section 3.1.

Line 182: "is converted and can be converted" – reword.

When the ice melts the surface lowers and the pressure drop in the liquid column of the hose is converted and can be converted into a surface lowering

When the ice melts, the hose coils up on the surface and the liquid column pressure drops and this drop in pressure is converted to surface lowering ($Z_{pta}$) by:

equation

Line 187: Unclear, what does mean "contains some noise", how does this relate to the 4 cm? Are the 4 cm independent of the pressure levels?

This uncertainty was erroneously based on the variability/standard deviation of the sensor prior to the onset of ice melt. During this period the hose is affected by the overlying pressure of the snow and thus the 4 cm is too high:

The uncertainty of the instrument is estimated to be 4 cm and contains some noise.

The accuracy of the pressure transducer is 2.5 cm and the standard deviation of the signal after the ice melt season has ended is 1.5 cm, with no systematical change relating to the depth of the sensor.

Line 188: how is the start of melt defined?

For the purpose of making the data easy to use the ice ablation observation is set to zero at the beginning of every melt season. This is done by subtracting the mean of a week prior to the onset of ice melt.

For the purpose of making the data easy to use the ice ablation observation is set to zero at the beginning of every melt season. This is done by subtracting the mean of a week prior to the onset of ice melt. The onset of ice melt is defined manually for each year by combining albedo, $Z_{pta}$ and $Z_{boom}$.

Lines 196-197: "should be in a state to be used directly for the continued monitoring of the A. P. Olsen transect." Reword, unclear also with added explanations in the following sentence.

Quality control is done to the best of our current knowledge, but the data is considered living data and should be in a state to be used directly for the continued monitoring of the A. P. Olsen transect. This means that corrections and filtering of data might change in future versions of the dataset. The filtered data could offer significant insights, and this is therefore included as supplementary data.

Corrections and quality control of the data is done to the best of our current knowledge, but the dataset is considered living data and should be directly comparable with data from the continued monitoring at the A. P. Olsen transect. As an example this means that if a better method for correcting the radiation sensor for tilt is implemented in the continued monitoring, the dataset will be updated to ensure consistency. The unfiltered data could offer significant insights, and this is therefore included as supplementary data.

Line 198: "and this is" should this read "and are therefore included"? Even if so, I do not fully understand what is meant.

I think it got unclear because it said filtered data where it should be unfiltered data (see also above)

The filtered data could offer significant insights, and this is therefore included as supplementary data.

The unfiltered data could offer significant insights, and this is therefore included as supplementary data.

Figure 4: As with other figures or tables, this item appears before it is referenced in the text.

I am not sure why this happens, but I have checked the latex file and the figures are inserted immediately after the first paragraph they are cited in. I hope the editorial office can make sure that this will be correct in the final version.

Line 244/245: "after snow cover" What is meant, after melting of the snow cover? "… non-tilt corrected data could potentially provide information discerning cloud cover variations, but using this should be approached with caution as absolute values are not reliable." I do not understand. Is this relevant? Consider removing or reword.

I agree that the part about the non-tilt corrected data is redundant and has been removed.

In January 2020, a shift in $Tilt_y$ at ZAC_L occurred, from field notes this can be explained by damage to the tripod legs and following loosening of the guy wires, after snow cover. The non-tilt corrected data could potentially provide information discerning cloud cover variations, but using this should be approached with caution as absolute values are not reliable.

In January 2020, a shift in $Tilt_y$ at ZAC_L occurred, from field notes this can be explained by damage to the tripod legs and following loosening of the guy wires, after the station being covered with snow.

Lines 251/252: The years cannot be corrected, consider rewording, making clear what exactly is corrected.

Specifically at ZAC_L the years spanning 2012 to 2016 and 2018 to 2020 needed to be corrected more than other years, and uncertainty is expected to be higher for these years.

Specifically at ZAC_L incoming shortwave radiation from the years spanning 2012 to 2016 and 2018 to 2020 needed to be corrected more than in other years, and uncertainty on $SR_{in}$ is expected to be higher for these years.

Lines 261/262: Add a very brief justification for these thresholds. Have these thresholds been used elsewhere? Literature sources?

There is no citable justification for the thresholds, and the paragraph is rewritten in order to explain that this is part of the QC filetering.

The incoming longwave radiation $LR_{in}$ is automatically filtered to remove data lower than 120 $Wm^{-2}$ and the outgoing longwave radiation $LR_{out}$) is filtered to remove data lower than 150 $Wm^{-2}$. The outliers are believed to occur on riming events and choice of limits is based on a visual assessment of outliers (Figure 10).

The incoming and outgoing longwave radiation shows some instances of outliers of unusual low values. We believe these events are caused by riming events. The most extreme cases are filtered out by excluding all incoming longwave radiation data ($Lr_{in}$) lower than 120 $Wm^{-2}$, and all outgoing longwave radiation ($LR_{out}$) lower than 150 $Wm^{-2}$ (Figure 10).

Figure 8: Top of atmosphere radiation is shown as a line in the legend but seems to refer to the grey background shading. How is this to be interpreted? I understand the upper threshold but not the lower one.

The span of the gray background shading is the minimum and the maximum theoretical top of atmosphere radiation, and thus the bounds are due to the daily cycle. The description of the figure in the text (see below) as well as the figure caption is updated.

Figure 8 caption:

ZAC_L: Daily maximum and minimum values of incoming shortwave radiation, corrected and uncorrected for tilt compared with top of atmosphere irradiance. Panel (a) is 2009 and panel (b) is 2016

Assessment of the effect of tilt correction (Panel a) and 2016 (Panel b) at ZAC_L: The shaded gray area span the daily calculated maximum and minimum top of atmosphere incoming short wave radiation (see equation 6). Solid lines represents the daily maximum observed incoming shortwave radiation before (gray) and after (yellow) the tilt correction . Similarly, the dashed lines represent the daily minimum observed radiation before (gray) and after (yellow) the tilt correction.

From the original manuscript line: 246-253 including the corrections from further above:

Thus, in order to evaluate the success of the tilt-correction and the quality/uncertainty of the radiation data we compare corrected and non-corrected shortwave incoming radiation in Figure 8. The top of atmosphere irradiance ($I_{toa}$, Equation 8) is used as a visual guideline in the comparison. Panel (a) in Figure 8 with data from ZAC_L in 2009, shows a successful year where the tilt correction modifies the values slightly. Panel (b) in Figure 8 shows a year where the tilt of the station has been more severe and uncertainties must be assumed higher in such years. Specifically at ZAC_L incoming shortwave radiation from the years spanning 2012 to 2016 and 2018 to 2020 needed to be corrected more than in other years, and uncertainty on $SR_{in}$ is expected to be higher for these years. Figure 8 also shows the minimum values of observed $SR_{in}$ are ranging below the minimum $I_{toa}$, indicating a substantial diffuse component.

Thus, in order to evaluate the success of the tilt-correction and the quality/uncertainty of the radiation data we compare corrected and non-corrected shortwave incoming radiation in Figure 8. The top of atmosphere irradiance ($I_{toa}$, Equation 6) is used as a visual guideline where the shaded gray area shows the span of $I_{toa}$ over a day. Panel (a) in Figure 8 with data from ZAC_L in 2009, shows a successful year where the tilt correction modifies the values slightly. Panel (b) in Figure 8 shows a year where the tilt of the station has been more severe and uncertainties must be assumed higher in such years. Specifically at ZAC_L incoming shortwave radiation from the years spanning 2012 to 2016 and 2018 to 2020 needed to be corrected more than in other years, and uncertainty on $SR_{in}$ is expected to be higher for these years. Figure 8 also shows the minimum values of observed $SR_{in}$ are ranging well below the minimum $I_{toa}$, this is due to the shading of the station in particular during summer nights when the sun angle is low and coming from north.

Figure 11, x-axis: While described, the axis is confusing. Could this be improved, showing ticks for the start of the individual months and labelling at least one of the months for every year (e.g. 1-7-2011 instead of only 2017)? Maybe a discrete vertical grid would improve readability of the figure.

We agree, and have made a figure with subplots in stead to make it clearer.

Line 285: snow -> snowfall

Corrected

Line 290: Snow-> Snowfall; is -> are

Snow events during the ice melt season is not captured by the PTA.

Snowfall events during the ice melt season are not captured by the PTA.

Lines 294/295: "how a point energy budget melt model is evaluated with the observed ice ablation." Is the model evaluated? Or are the data evaluated using the model? This might be somewhat of an open question. The abstract suggests rather that the model is used to evaluate the data ("We showed that the **inherent uncertainties of the data** resulted in an accurate reproduction of ice ablation for just one of the two years."), while here you state the opposite.

We will exemplify here how a point energy budget melt model is evaluated with the observed ice ablation.

In this use case we exemplify how a point energy budget melt model can be set up using the observed variables.

Line 303: It might be fine to neglect G, but this needs to be justified, at least to a minimal degree, e.g. citation of relevant literature. Same for sublimation further below.

For the purpose of this example we neglect G and calculate the turbulent heat fluxes following Monin-Obukhov theory […]

For the purpose of this example we neglect G assuming the contribution from this is minor to the contribution from other sources as in Abermann et al., (2019). The turbulent heat fluxes are calculated following Monin-Obukhov theory […]

With regards to sublimation, we think that it is actually not necessary to account for sublimation for a melting surface only and thus the statement is redundant:

This is only valid for a melting surface and sublimation is not accounted for.

This is only valid for a melting surface.

Line 320, citation: Broeke -> van den Broeke.

Corrected

Line 322: somewhat unclear, maybe instead "day n-1 to day n"?

When double checking this we found a descrepancy between what was actually done in the code and what was written. The paragraph is clarified but content is therefore slightly changed.

The performance of the melt model on a daily timescale is evaluated by summing up modeled melt to daily values and taking the difference between the daily mean $Z_{pta}$ value from the day before to the current day to get observed daily values (Figure 12 panel (a) and (b))

The performance of the melt model on a daily timescale is evaluated by summing up modeled melt to daily values and comparing these to the observed daily melt rates. The observed melt rates are calculates as the difference between the daily minimum and the maximum value of the $Z_{pta}$ (Figure 12 panel (a) and (b)).

Line 333: accumulated -> cumulative

Corrected

Line 325: "…, while the model performs poorly in 2016 for all three roughness factors." This statement cannot remain as it is. I agree that none of the three match, but the three $z_{0w}$ values do define a range of uncertainty in the parameter, not the only three discrete and possible values. Hence, the question is not whether one of the three calculated melt curves coincides with the measurements, the question is whether the measurements fall within the range of the three simulated melt curves. This is the case for both years.

See collected answer below

Line 327/328: "In conclusion…" please reword, consider making two sentences.

See collected answer below

Line 331: "all major components" This is not correct as the data do not include the turbulent fluxes which are a key component of the energy balance. They can be calculated from the measurements, hence slightly reformulate: The dataset does not comprise the components of surface energy balance directly, but all meteorological parameters relevant to calculate the surface energy balance.

See collected answer below

Conclusions of Section 5; interpretation of the results from Section 5 in the abstract: I do not agree to these conclusions. I understand that this is a data paper and that the use case should be kept short. However, some more discussion is needed. In both years, the measured melt lies well within the range of model output. At the same time,

the model relies on various simplifications and there is no appropriate uncertainty calculation, neither in model output, nor in measured melt. The coinciding value for $z_{0w}$ =0.001 in 2009 could be a coincidence. How good is the agreement that one typically expects from an energy balance model, consider citing other studies? Is there a reason that you mainly look at the results from $z_{0w}$ = 0.001? If yes, please justify.

I do not expect the authors to carry out a complete uncertainty analysis like, e.g., Machguth et al. (2008) or Zolles et al. (2019). However, uncertainties in energy balance modelling are considerable and even without any complete uncertainty analysis, modelled and simulated melt overlap in this case. Hence, I am not sure that this analysis tells anything else than that model and measurements agree within their simple bounds of uncertainty (simple referring to the fact that only one parameter was varied within bounds of uncertainty).

We agree on the comments above, and the purpose of the use case was in the first place not to evaluate on the data quality – as the reviewer is rightfully pointing out this would require much more rigorous uncertainty analysis which is out of the scope of the manuscript. For this reason we revise the manuscript to present the use case as what it is, a simple use case as an example of how the data can be implemented.

We start by rewording the abstract slightly:

The usability and some quality issues are exemplified by using the data in an energy balance melt model for two different years. We show that the inherent uncertainties of the data result in an accurate reproduction of ice ablation for just one of the two years.

The usability of the data is exemplified by using the data in an energy balance melt model for two different years.

From the first sentence in the use case section:

The variables collected at the A. P. Olsen transect are key variables in the surface energy budget equations, and can be used for calculating the energy availability for melting ice. In this use case we exemplify how a point energy budget melt model can be set up using the observed variables and use this to evaluate the usability and quality of the data set.

The variables collected at the A. P. Olsen transect are key variables in the surface energy budget equations, and can be used for calculating the energy availability for melting ice. In this use case we exemplify how a point energy budget melt model can be set up using the observed variables.

The second to last paragraph in the use case section:

The point melt is computed for three values of the surface roughness factor for wind, $z_{0w}$: 0.01, 0.001 and 0.0001, as this value has been shown to vary with orders of magnitude (e.g. Smeets and van den Broeke, 2008). The performance of the melt model on a daily timescale is evaluated by summing up modeled melt to daily values

and comparing these to the observed daily melt rates. The observed melt rates are calculates as the difference between the daily minimum and the maximum value of the $Z_{pta}$ (Figure 12 panel (a) and (b)). The model performance in 2009 shows good agreement with the observed ablation when using $z_{0w}$=0.001, while the model performs poorly in 2016 for all three roughness factors. This could indicate that the quality of the radiation data is questionable in 2016 and/or that the ice ablation data is spurious.

The point melt is calibrated by varying the surface roughness factor for wind, $z_{0w}$ within a range between 0.01 and 0.0001, as this value has been shown to vary with orders of magnitude (e.g. Smeets and van den Broeke, 2008). All the uncertainties introduced by both model assumptions are in this way summarized in this single static value. For the purpose of this example we define a successful calibration on a seasonal scale thus choosing the value of $z_{0w}$ that gives a total melt over a melt season that best match the total observed ablation over the same season (Figure 12 panel (a) and (b)). Model performance is then evaluated on daily timescale by accumulating the modeled melt to daily sums and comparing these to the observed daily melt rates (Figure 12 panel (b) and (c)). The observed melt rates are calculated as the difference between the minimum and the maximum value of the $Z_{pta}$ over a day. A value of $z_{0w}$=0.001 was found to match the 2009 total ablation, while $z_{0w}$=0.005 was more appropriate for 2016. The performance of the melt model on a daily scale is affected by both model assumptions as well as observational uncertainty, and we believe that the poorer performance in 2016 is related to the higher uncertainty in the shortwave radiation observations due to a high tilt of the station and/or unknown issues with the ablation sensor.

From the conclusion:

The dataset contains all major components of the surface energy balance

The dataset contains key components to calculate the surface energy balance

**Cited literature**

Machguth, H., Purves, R. S., Oerlemans, J., Hölzle, M., & Paul, F. (2008). Exploring uncertainty in glacier mass balance modelling with Monte Carlo simulation. The Cryosphere, **2,** 191–204. https://doi.org/10.5194/tc-2-191-2008

Zolles, T., Maussion, F., Galos, S. P., Gurgiser, W., & Nicholson, L. (2019). Robust uncertainty assessment of the spatio-temporal transferability of glacier mass and energy balance models. The Cryosphere, **13**(2), 469–489. https://doi.org/10.5194/tc-13-469-2019

**Referee #2:**

The reviewed manuscript establishes a quality-assured near-surface climate and ablation data collected by three Automatic Ablation and Weather Stations in northern Greenland. This manuscript focuses on the quality of the data and its usability in the study of ice ablation processes, emphasizing the unique value of this long-term dataset in understanding the response of peripheral glaciers to climate change.

In principle, this work is important as it provides a very valuable long-term series monitoring dataset of Greenland's peripheral glaciers which are sensitive to climate warming. The manuscript detailly outlines the quality control procedures, assesses data integrity, and provides explanations for data gaps. The dataset of this study would be of interest to the community of geomorphology and cryosphere stakeholders, especially in the context of modern climate change. Overall, I find the paper to be well-organized with clear logic.

My main concern is the readability of the articles, as they can be difficult for readers to follow. Particularly troubling are the diagrams within the manuscript, which lack clear legends and can consequently confuse readers. Additionally, the figure captions are too brief and fail to provide a detailed description of the figures they represent. Furthermore, I have concerns regarding the representativeness of the chosen sampling points. To address this, I suggest including an explanation that details the rationale behind the selection of these specific points.

Overall, I think this is a nice and important work that is suitable for ESSD. I would recommend a minor revision.

Thank you for constructive and helpful comments, they are highly appreciated. We agree to all concerns and the manuscript has been improved by implementing the suggestion. Please see line by line answers below.

Major comment:

While Figure 1 shows the locations of the three AAWSs on the A.P. Olsen ice cap, and Table 1 provides their coordinates, readers may be interested in the broader context of the sites' location within Greenland. It is suggested to add an inset map of Greenland in Figure 1 to show the distribution of the sites.

The map has been updated with an insert of Greenland.

The melting of marginal glaciers is a major contributor to sea level rise. In addition, what are the other important ecological and social implications of marginal glaciers compared to the Greenland ice sheet? It is suggested to add this in the first paragraph of the introduction to highlight the importance of this study.

Yes, very important to mention. Therefore we include a sentence in the first paragraph of the introduction:

Under the influence of the current warming climate, glaciers and ice caps exhibit a pronounced negative surface mass balance, contributing to sea level rise. Perhaps equally important are the local scale changes occurring in glaciated catchments where the volume and timing [...]

Under the influence of the current warming climate, glaciers and ice caps exhibit a pronounced negative surface mass balance, contributing to sea level rise. Ice loss from glaciers distinct from the Greenland Ice Sheet are on a par with the mass loss of the ice sheet. Globally, the melting of glaciers distinct from the main ice sheets accounts for approximately 25-30% of the sea level rise attributed to the melting of land ice. Perhaps equally important are the local scale changes occurring in glaciated catchments where the volume and timing [...]

Lines 211-214: A more logical explanation is needed here as to why the inability to recharge a battery does not affect temperature observations during the winter months.

This most often happens during winter when the batteries cannot be re-charges due to the polar night, coinciding with the period where ventilation of the casing is not necessary.

This most often happens during winter when the batteries cannot be re-charged due to the polar night, coinciding with the period where ventilation of the casing is less important as the casing is not heated by shortwave radiation.

Minor comments:

Figure 4: Differences in gradients between sites need to be added to the figure caption in more detail, such as the difference between gray and red curves.

Air temperature quality control. Panel (a) and (b): Unfiltered and filtered data respectively, ZAC_L is blue, ZAC_U is orange and ZAC_A is green. Panel (c): The temperature gradient per 100 m between ZAC_L and ZAC_U, ZAC_U and ZAC_A, ZAC_L and ZAC_A respectively.

Air temperature quality control. Panel (a) and (b): Unfiltered and filtered data respectively, ZAC_L is blue, ZAC_U is orange and ZAC_A is green. Panel (c): The temperature gradient per 100 m between ZAC_L and ZAC_U, ZAC_U and ZAC_A, ZAC_L and ZAC_A respectively. Gray is data considered to show natural variation and red is flagged data considered to show variability caused by a faulty sensor at one of the stations.

Figure 11: The $Z_{stake}$-curve of 2013 is significantly different from other years. There is a suspected case of multiple m ice values corresponding to one date, and a reasonable explanation needs to be given.

True, for some reason we skipped the explanation of 2013. A sentence is added in the text.

This indicates no consistent under-catch in the PTA system. In 2012, the melt rates of both systems were similar until late July. The discrepancy might be due to another

collapse of the sonic ranger stake assembly. For 2015 and 2016 the melt rates were closely aligned between the two systems.

This indicates no consistent under-catch in the PTA system. In 2012, the melt rates of both systems were similar until late July. The glacier melt in 2013 was the highest on the A. P. Olsen record, the variability in sonic ranger observations in particular late in the season could suggest the stake system being almost melted out and unstable. For 2015 and 2016 the melt rates were closely aligned between the two systems.

Figure 12: Resolution is significantly lower than the other figures.

The figure has been updated, also with better resolution.

Line 226: How to assess whether data has been affected.

The cause of the drift in relative humidity values from 2016 at ZAC_A is unknown, and the affected data has been discarded.

The cause of the drift in relative humidity values from 2016 at ZAC_A is unknown, and the data up until the replacement of the HygroClip (which coincides with the end of the record presented here) been discarded.

Section 5 shows a successful application of the point energy budget melt model. But in Figure 12, a and b, the deviation of the simulated values from the observed values is not given. It is suggested to add the specific values for the quality assessment.

Section 5 has been revised according to this comment as well as comments from the other reviewer.

---

## Author Response (AR3)

Dear Editor,

We thank both reviewers for taking the time to go through the manuscript once again. We have addressed all comments and have gone through the entire manuscript correcting for readability errors once again.
Referee #1 questions the use case in section 5 should be part of the main text or in an appendix. Both reviewers from the first round were happy with it, but we will leave it up to the editor to decide if this section is appropriate for an ESSD data paper.

Thank you for considering this dataset and paper for publication.

Best regards,
Signe Hillerup Larsen

Report #2.

Review of "Climate and ablation observations from automatic ablation and weather stations at A. P. Olsen Ice Cap transect, NE Greenland, May 2008 through May 2022" by Signe Hillerup Larsen et al.

Earth System Science Data #essd-2023-444

General comments:

Please note that this reviewer did not participate in the open discussion; therefore, this reviewer only evaluates the authors' responses to the initial reviewers' comments/suggestions. The line numbers below are from the authors' responses (version 3). My overall impression is that the authors' responses are generally poor: In most of the responses, they indicate only the original and revised texts, and therefore, it is frequently unclear whether they agree or disagree with each review comment, which makes it difficult for this reviewer to assess the authors' responses appropriately. However, this reviewer suggests that this data paper can be considered for publication in the journal ESSD once the authors attend to the following comments/suggestions.
We thank the reviewer for going through our answers to reviewers and apologies for not giving more details in each answer. We agree to all reviewer comments and please see answers to the specific comments below.

Specific comments

Referee #1:
Lines 57-62: I think the referee suggests that "in-situ observations of near surface climate and ablation are available from very few peripheral glaciers distinct from the Greenland ice sheet in Greenland, and a transect of three AAWSs is, to the current knowledge of the authors, unique to Greenland." can be merged with the second paragraph of the introduction section. In my opinion, the authors did not answer this comment sufficiently. Please consider the suggestion again.
We agree the part about the importance of the in-situ observations is already stated in the second paragraph of the introduction and the uniqueness of the transect of three stations has now been placed up in the beginning of the third paragraph.

Lines 82-85, line 90, and Line 92-93: The authors' response is unclear. What did they do in response to the referee's comments? Because these points are related to the major comments, the authors must attend to them carefully and describe their responses in detail.
It is correct, we should have responded to these points, but didn't.

The details in lines 82-85 has been moved to the data availability section.
The comment for line 90, where it is stated that ice ablation is mentioned in the head line but is not described in the section, has been addressed by the restructuring of the text and headlines match the content now.

Regarding line 92-93. The paragraph has been rephrased in order to make it clearer to the reader how height of the instruments relative to the surface can change over the year.

Lines 118/119: Table 3 has been updated in response to the referee's comment. However, the intention of

"maintenance" is unclear. Do the authors intend to indicate information on the sensor replacement schedule using the word? I think using another word instead of "maintenance" is better.
We agree to this point and has changed the column to: Calibration.

Line 168: Regarding the height of the installation, it is informative for readers if this height information is summarized in Table 3.
Good idea. The height above surface is now in Table 3.

Line 188: The authors' response here is not enough. What do the authors do by combining albedo, Z_pta and Z_boom to detect the melt onset manually? This is the most critical point here.
We agree that this was still quite unclear. We have elaborated the description of how we, manually, find the onset of melt.

Lines 196-197: The revised sentences are still redundant, in my humble opinion. Suggest rephrasing "Corrections and quality control of the data is done to the best of our current knowledge, but the dataset is considered living data and should be directly comparable with data from the continued monitoring at the A. P. Olsen transect. As an example this means that if a better method for correcting the radiation sensor for tilt is implemented in the continued monitoring, the dataset will be updated to ensure consistency. The unfiltered data could offer significant insights, and this is therefore included as supplementary data." -> "The unfiltered data are included in the dataset because it allows us to update it when a better quality-control method is developed."
We agree that there is some redundancy in this. We have revised this part of the text and now it just says: "The unfiltered data could offer significant insights to expert users and are thus included as supplementary data in the dataset."

Figure 8: Although the caption has been updated to include an explanation of the shaded gray area, the figure legend still includes a gray line plot (I_toa) that should be modified as well.
The legend is correct, it is just that due to the high variability of the I_toa due to the daily solar cycle, the line looks like a shaded area. We have made the line thinner and the figure 2 columns wide in order to make it clearer what the line represents.

New section 5, "Use case": In my opinion, this section is not necessary for the main content at all because its purpose is to show the energy balance model's sensitivity to the parameter choices of roughness lengths for momentum and its accuracy in terms of daily melt amount, which are not relevant to the data itself. Please consider removing the section from the main content and moving it to supplementary material.
This has also been discussed amongst the authors, however, since the two other reviewers have been happy with this, we will leave it up to the editor to decide if the section does not match the format of ESSD.

Referee #2:
Major comment 1: Although the authors have updated Fig. 1 following the referee's suggestion, some figure descriptions are still difficult to follow. The figure legend tells that the glacier catchments are indicated with black and gray outlines; however, they are difficult to identify. Additional explanations are needed. In the "base map," a black box is indicated. Does it indicate the exact area shown in the main figure? It should be explained in the caption.
We agree that it is hard to see the exact outlines. The figure has been updated, and the overview map legend and black box has been altered to cover the exact map area and the outlines have been made more visible.

Major comment 2: Regarding the newly added sentence, "Ice loss from glaciers distinct from the Greenland Ice Sheet are on a par with the mass loss of the ice sheet. Globally, the melting of glaciers distinct from the main ice sheets accounts for approximately 25-30% of the sea level rise attributed to the melting of land ice." a reference for the statement is needed.
This sentence has been revised and the text changed in order to cite a more recent source that was used in earlier versions of this manuscript:
"Peripheral glaciers and ice caps (GICs) that are separate from the Greenland Ice Sheet make up only about 4% of Greenland's total glaciated area but are responsible for approximately 14% of the island's current ice loss, contributing disproportionately to the overall ice reduction (Khan et al., 2022)."

Figure 4 caption: Suggest rephrasing "Gray is data considered to show natural variation and red is flagged data considered to show variability caused by a faulty sensor at one of the stations." -> "Gray line indicates

data considered to show natural variation and red line denotes flagged data considered to show variability caused by a faulty sensor at one of the stations."
Thank you for this concrete suggestion, we have changed the caption according to the suggestion.

Figure 11: Why did the authors remove the figure legend incorporated in the original figure? This is certainly necessary. For the 2013 rapid decrease in the sonic ranger observation, the authors state that the decrease was caused by the situation that "the stake system being almost melted out and unstable". If it is true, the rapid reduction in 2013 does not tell a realistic state. Please consider masking the rapid reduction.
A legend has been inserted and the SR50 data from 2013 has been removed completely, as we agree that it makes no sense to compare with data we do not trust to be good.

L. 226: In my humble opinion, the authors' quality control of relative humidity is subjective. It seems to me that the authors think that the relative humidity values from ZAC_L and ZAC_U are more reliable than those from ZAC_A. How do the authors find that? A more objective method should be considered and applied.
The quality control of relative humidity is really hard to do objectively. And it is correct that perhaps we have been biased towards trusting the lower stations more. Investigating trends in median values, we could not define any objective method for rejecting the data and we will based on this discard less data for ZAC_A.

The paragraph has been changed, and the data has been updated to include former discarded data:
"The humidity sensor typically requires recalibration every 1-2 years. However, due to logistical challenges, this was not always feasible, and an uncalibrated sensor will drift towards increasingly poorer performance. Drifting values of relative humidity are hard to objectively quantify. From a visual inspection of the relative humidity time series in Figure 5, drifting could have occurred at ZAC_A during 2012-2014 and we will leave it up to the user to define when data is useful."

**Report #1**

Summary and major points

I thank the authors for their revisions which address the points raised in my previous review. I do only have a few minor points to raise concerning the revised version of the manuscript. In general, I had the impression that the revisions were done a bit quickly here and there, with words sometimes missing and other minor inconsistencies being introduced. I list a few of these issues below, but there might be more that i have not noticed. I suggest to thoroughly scan the document once again before submitting the final version.
We thank the reviewer for going through the manuscript once again and for the suggested corrections. We have gone through the entire manuscript to correct for readability errors, missing commas etc.  Please see the line by line answers below.

Detailed remarks

Line 20: I think it should read "is on a par". Furthermore, I am surprised by this statement as it suggests that total mass loss from the ice sheet and from the local glaciers is very similar, which I doubt. Is this statement correct? Please modify so this is clearer and in any case provide a citation.
This sentence has been revised and the text changed in order to cite a more recent source that was used in earlier versions of this manuscript:
"Peripheral glaciers and ice caps (GICs) that are separate from the Greenland Ice Sheet make up only about 4% of Greenland's total glaciated area but are responsible for approximately 14% of the island's current ice loss, contributing disproportionately to the overall ice reduction (Khan et al., 2022)."

Lines 21 - 22: What are the sources of these numbers? Please provide citations.
This has been addressed by the above revisions.

Line 31: "This all sums up…" or "These all sum up …"
The sentence has been rephrased to "This all sums up…"

Lines 113-114: The statement is unclear, sounds like the HygroClip has been replaced with some other type of instrument. I assume you mean that the HygroClips are regularly (what interval?) replaced with a freshly calibrated HygroClip.
The sentence has been slightly restructured to make it clearer:
"The HygroClip has been replaced with a freshly calibrated instrument at each field visit."

Line 232: This is the second time you use the title "Ice ablation", please use another heading, for example "Correction of measured ice ablation".

Thanks for the concrete suggestion the section title is changed to "Correction of measured ice ablation".

Line 287: A verb is lacking here: "...) been discarded." Insert "have" or "has", depending on your use of the word "data".
This sentence has been removed completely based on a comment from the other reviewer.

Figure 8: The dashed lines for minimum daily radiation are very hard to read. I suggest to make better use of the space in the figure and set the maximum of the y-axis to around 900 w m-2?
Thanks for this concrete suggestion, the figure has been updated both with the new y-lim at 900 and it has also been made broader to be able to clearly distinguish all lines.

Figure 8, caption, first sentence: The year is not mentioned for panel a but for panel b.
Thanks for noticing this error, it has been corrected so that it is clear that panel a is 2009.

Figure 11: Which colour denotes which instrument?
A legend has been inserted and reference to the color has been added in the caption.